# A Case of Persistent Diarrhea in a Man with the Molecular Detection of Various *Campylobacter* species and the First Isolation of *candidatus*
*Campylobacter infans*

**DOI:** 10.3390/pathogens9121003

**Published:** 2020-11-30

**Authors:** Jacky Flipse, Birgitta Duim, Janny A. Wallinga, Laetitia R. H. de Wijkerslooth, Linda van der Graaf-van Bloois, Arjen J. Timmerman, Aldert L. Zomer, Kees T. Veldman, Jaap A. Wagenaar, Peter Bloembergen

**Affiliations:** 1Laboratory of Medical Microbiology and Infectious Diseases, Isala Clinics, 8025 AB Zwolle, The Netherlands; j.wallinga@certe.nl (J.A.W.); p.bloembergen@isala.nl (P.B.); 2Department of Infectious Diseases and Immunology, Faculty of Veterinary Medicine, Utrecht University, 3584 CL Utrecht, The Netherlands; l.vandergraaf@uu.nl (L.v.d.G.-v.B.); a.j.timmerman@uu.nl (A.J.T.); a.l.zomer@uu.nl (A.L.Z.); j.wagenaar@uu.nl (J.A.W.); 3WHO Collaborating Center for Campylobacter/OIE Reference Laboratory for Campylobacteriosis, 3584 CL Utrecht, The Netherlands; 4Department of Gastroenterology & Hepatology, Isala Clinics, 8025 AB Zwolle, The Netherlands; l.r.h.de.wijkerslooth@isala.nl; 5Wageningen Bioveterinary Research, 8221 RA Lelystad, The Netherlands; kees.veldman@wur.nl

**Keywords:** *Campylobacter* spp., non-jejuni/coli infection, culture versus PCR

## Abstract

A man with a well-controlled HIV infection, previously diagnosed with lymphogranuloma venereum and treated for Hodgkin’s lymphoma, was suffering from chronic diarrhea. He travelled to Indonesia in the month prior to the start of complaints. Over a 15-month period, sequences related to *Campylobacter*
*troglodytis/upsaliensis*, *C. pinnepediorum/mucosalis/concisus* and *C. hominis* were detected by 16S rRNA qPCR-based assays in various stool samples and in a colon biopsy. Culture revealed the first isolation of “*candidatus Campylobacter infans*”, a species identified recently by molecular methods only. The patient was treated with azithromycin, ciprofloxacin and tetracycline. To identify potential continuous exposure of the patient to *Campylobacter*, stool samples of the partner and the cat of the patient were analyzed and *C. pinnepediorum/mucosalis/concisus* and *C. helveticus*, respectively, were detected. The diversity in detected species in this immunocompromised patient with a lack of repeatedly consistent findings resulted in the conclusion that not any of the *Campylobacter* species was the primary cause of the clinical condition. This study shows the challenges in detection and interpretation of diagnostic results regarding *Campylobacter*.

## 1. Introduction

The *Campylobacter* genus was recognized in 1963 and includes several important human and animal pathogens. There are currently 33 documented *Campylobacter* species (as of September 2020, http://www.bacterio.net/campylobacter.html), some of them being associated with infections in humans. The most frequently reported *Campylobacter* species in humans are *Campylobacter jejuni* and *C. coli.* These species are considered a major cause of gastroenteritis world-wide [1]. In humans, *Campylobacter* species have been associated with gastrointestinal infections, including inflammatory bowel diseases, Barrett’s esophagus and colorectal cancer, but also other sporadic clinical outcomes such as bacteremia, meningitis, pneumonia, brain abscesses and reactive arthritis [2,3]. A considerable proportion of the non-jejuni/coli infections occur in immunocompromised humans [4]. Culture-based detection of intestinal *Campylobacter* species is performed using selective media. Most of these selective media are developed and optimized for the detection of *C. jejuni* and *C. coli*. The detection of non-jejuni/coli species in stools is hampered by the susceptibility of these species to antimicrobials used in routine jejuni/coli media and the fastidious growth of many of the other *Campylobacter* species [5,6]. To overcome the problem of growth-inhibition by antimicrobials, the filter method can be used, which employs a 0.6 µm filter on blood agar to select for motile bacteria. However, this method has a higher detection limit. Alternatively, molecular methods can be used but not all diagnostic laboratories have established these assays. Moreover, the currently used assays have their limitations in sensitivity and specificity, depending on the assay used. Finally, the causative role of several *Campylobacter* species in intestinal disorders has not been established in humans yet. Here, we present a case report of a patient that suffered from chronic gastroenteritis. This study shows the challenges in detection and interpretation of diagnostic results. This study describes the first isolation of *candidatus Campylobacter infans*, a novel species that has been identified recently by molecular methods only [7].

## 2. Case Report

A 42-year old man presented himself at the outpatient clinic with complaints of chronic diarrhea and abdominal discomfort. He had a history of well-controlled HIV infection (viral load < 20 IU/mL), lymphogranuloma venereum and treated Hodgkin’s lymphoma, which was stable at the time of complaints. He reported a trip to Indonesia two months prior to visiting the outpatient clinic and was considered to be at risk for a gastrointestinal infection (Figure 1 represents the timeline with the travel to Indonesia at month 0 and visit to outpatient clinic at month 2).

As part of the clinical evaluation, a stool sample was taken and tested for gastrointestinal pathogens. The stool sample (month 2) tested positive using an in-house qPCR for *Campylobacter* species [8], which targeted 108 base pairs (bp) of the 16S rRNA gene. The stool sample was further tested by qPCR for multiple gastroenteric pathogens using the following targets: toxigenic *Clostridioides difficile* (*tcdA*), *Salmonella* spp. (*ttr*), *Yersinia enterocolitica* (*Yst*), enteroinvasive *Escherichia coli* or *Shigella* spp. (*ipaH*), enteropathogenic *E. coli* (*eaeA*), enterotoxic *E. coli* (*St*, *Lt*, *Stx1* or *Stx2*), *Giardia lamblia* (*18S*), *Entamoeba histolytica* (*18S*), *Cryptosporidium parvum/hominis* (*DNAJ-like*), norovirus (*ORF1/ORF2*), rotavirus (*NSP3*), adenovirus (*Hexon*), sapovirus (*orf1*) and astrovirus (*Orf1a*) ([8]; unpublished). All these qPCR targets were negative. Microscopy did not identify parasites in the sample. The stool sample was sent to external reference laboratories who reported negative qPCRs for *Tropheryma whipplei* and microsporidia (*Enterocytozoon bineusi*, *Encephalitozoon* spp.).

All qPCR-positive stools were plated on Karmali agar (Oxoid Ltd., Hampshire, UK) to select for *Campylobacter* species. Liquid or resuspended stools were plated and incubated for three days under microaerobic conditions at 42 °C. After month 10, culture protocols were extended for this patient with Skirrow-agar and Belo horizonte medium agar (MediaProducts, Groningen, The Netherlands) [9,10]. Skirrow-agar and Belo horizonte medium were incubated at 35 °C under microaerobic conditions, while growth was assessed every 2nd day for 6 days total. None of the media resulted in *Campylobacter*-positive cultures.

Based on the positive qPCR for *Campylobacter*, the patient was treated for three days with azithromycin (once daily 500 mg for 3 days) [11,12]. However, gastrointestinal complaints persisted.

During this period of persisting diarrhea, two additional stool samples (months 3 and 4) were found positive for *Campylobacter* species by qPCR and negative in the qPCRs for the pathogens listed before. Because of ongoing complaints, a colon biopsy was performed at month 5. No pathological abnormalities were found in the biopsy. The 16S-qPCR, as described above, was positive, while the in-house-adapted species-specific qPCRs for *C. jejuni*, *C. coli*, *C. fetus*, *C. hyointestinalis*, *C. upsaliensis* and *C. lari* (Appendix A, [13,14]) were negative, indicating that potentially another *Campylobacter* species was involved. Subsequently, 470bp of the *Campylobacter* 16S rRNA gene was Sanger sequenced as previously described [15]. The resulting sequence had 99% homology (4bp mismatches) with *C. troglodytis* type strain MIT 05-9157 (HQ864829.1) [16] and to lesser extent with *C. upsaliensis* (8bp mismatches) (Figure 2). The latter species could be excluded since the *C. upsaliensis*-specific qPCR was negative.

Based on this result, the patient was treated with ciprofloxacin (2 daily doses (dd) 500 mg, for 7 days) [11] and the next stool sample (month 6) was negative for *Campylobacter* species-qPCR (Figure 1).

At month 10, the patient returned to the outpatient clinic as he still suffered from chronic diarrhea. The stool sample was positive for *Campylobacter* 16S rDNA by qPCR. Based upon the response to the fluoroquinolone treatment at month 5, the patient received another course of ciprofloxacin (2dd 500 mg for 7 days). Stored (month 6, 10) and a fresh stool sample (month 11) were submitted to the WHO Collaborating Center for *Campylobacter* in Utrecht, the Netherlands for culture. Culture was performed as previously described with the filter method [17]. Only the fresh sample (month 11) yielded one single *Campylobacter*-suspected colony, which was sub-cultured on Columbia agar with 5% sheep blood and incubated under microaerobic conditions at 37 °C for 5 days. The 16S rRNA sequence of this colony showed 94% homology with the 16S rRNA gene of *C. hyointestinalis subsp. lawsonii* and whole genome sequencing of the isolate identified 99% homology to the recently identified novel *Campylobacter* spp. “*candidatus Campylobacter infans*” [7,18]. None of the 16S rRNA sequences that were detected in months 5, 11 and 12 showed homology with this *Campylobacter* species. 

Antimicrobial susceptibility testing for this strain was performed using broth microdilution according to CLSI guidelines (Vet06). The results were interpreted using EUCAST clinical breakpoints [21]. As clinical breakpoints were not available for this species, the clinical breakpoints for *C. jejuni* were used. The strain showed resistance to macrolides (minimal inhibitory concentration (MIC) erythromycin >128 mg/L) and fluoroquinolones (MIC ciprofloxacin 16 mg/L), and susceptibility for gentamicin (MIC 1 mg/L), streptomycin (MIC 2 mg/L) and tetracycline (MIC ≤ 0.5 mg/L). Therefore, a 14 day-treatment with tetracycline (4dd 250 mg for 2 weeks) was started combined with follow-up diagnostics (Figure 1). The stool remained qPCR-positive for *Campylobacter* species after treatment for 7 days with tetracycline, but was qPCR-negative after 14 days of tetracycline (month 12) and at month 13 (approximately 4 weeks after the start of the tetracycline treatment). However, 6 weeks after finishing tetracycline treatment, the patient had ongoing gastrointestinal complaints and a stool sample was again qPCR-positive for *Campylobacter* species (month 14). As the isolated strain was susceptible to gentamicin, an oral treatment with 4dd 80 mg gentamicin was prescribed but stopped within 2 days due to side effects [11,12].

In the follow-up, stool samples from the cat in the household (month 12) and the partner of the patient (month 13) were analyzed and from the patient multiple stool samples after the course of tetracycline (month 14) were analyzed by qPCR (Figure 1). All stool samples were positive for *Campylobacter* DNA. Sequencing of the stool sample from the patient resulted in a mixed 16S rDNA sequence, which was analyzed with Mixed Sequence Reader [22]. The mixed sequences were most similar to *C. hominis*. The sequence retrieved from the stool of the partner was distantly related to *C. pinnepediorum*, *C. mucosalis* and *C. concisus*, whereas the sequence of the stool sample of the cat (month 12) showed highest homology to *C. helveticus* (Figure 2).

Finally the patient was diagnosed with irritable bowel syndrome having excluded other causes. The patient received one course of metronidazole (3dd 500 mg for 2 weeks) to treat for spirochetes.

Taken together, sequencing PCR products of the stool samples identified various *Campylobacter* species over the course of time. No single *Campylobacter* species could be identified as a consistent inhabitant in the patient (Figure 1). *Campylobacter* spp. remained detectable by qPCR despite antimicrobial treatments. However, qPCR-negative episodes occurred as well and *candidatus C. infans* was only detected once. During the course of tetracycline, the patient kept a diary of his complaints and appearance of the stools. No clear correlation could be found between the results of the *Campylobacter* PCR and abdominal complaints.

The possibility that re-infection occurred from within the household was investigated. No homology was found between *Campylobacter*-positive sequences from the patient and his cat, however, there was a high homology between qPCR results of one sample from the patient and one sample of his partner collected at two different time points (patient: month 11 and partner: month 13, respectively).

The local medical ethical committee (Isala clinics) waived the requirement for informed consent based on anonymous description of the case. All patients at Isala clinics are informed of their right to (retrospectively) refuse consent to have their materials being reanalyzed for scientific purposes.

## 3. Discussion

This report describes the molecular detection of various *Campylobacter* species over time and the first isolation of “*candidatus Campylobacter infans”* in an (potentially) immunocompromised patient suffering from chronic diarrhea. Due to lack of antimicrobial susceptibility data for the molecularly detected species, empiric antimicrobial therapy was given. This case shows several challenges in *Campylobacter*-related disease: how to identify the causative species, how to get information about the antimicrobial susceptibility for treatment and how to evaluate the microbiological findings in relation to the clinical presentation. Finally, there is the possibility of continuous exposure from the environment or persistent carriage in the patient.

Diagnostics for samples from diarrheal patients is increasingly performed with molecular assays. The use of these assays has changed the view on the epidemiology of some pathogens [8]. The introduction of 16S-based assays to detect members of the *Campylobacter* genus allows the detection of previously underdiagnosed species. It also raises questions when (partial) 16S sequences are not fully similar to sequences in databases. In this study findings were *C. troglodytis*-/*C. upsaliensis*-like (detected twice in the patient) and *C. mucosalis/pinnepediorum* (detected in the patient once and in the partner of the patient). The presence of more than one *Campylobacter* species might lead to amplification of multiple 16S rDNA sequences, as was shown in this study (patient, month 14). Using the Mixed Sequence Reader *C. hominis* was indicated. Sequences not fully congruent may identify *Campylobacter* species that may have been unrecognized in the patient so far or might concern uncharacterized emerging pathogens. Besides the technical diagnostic issues about detection and identification, these findings also require critical consideration regarding the (causative) association of these microbiological findings with the clinical situation of the patient. In this study, the findings were initially considered related to the patient’s complaints. However, the variety of detected species led to doubts about the causative relation. This case study shows the power of the molecular assays but also the limitations. An additional problem of molecular diagnostics is the lack of antimicrobial susceptibility data. Despite several attempts, it was only possible to isolate *Campylobacter* from one of the samples (month 11) using the filter method. This method does not use selective plates with antimicrobials which might have been the cause of negative cultures for the other samples. Only one colony was found which suggests that this strain was not present in high numbers, although the filter method has a relatively low sensitivity [17]. Remarkably, the isolated *Campylobacter* appeared to be the first isolate of a recently “constructed” potential new species based upon metagenomics data of stool samples of breast fed children with diarrhea from several sub-Saharan African and South Asian countries [7,23]. There is limited information about the epidemiology of this new species and the source of this strain for the patient is unknown. Therefore, the clinical relevance of this strain for the patient remains unclear. It might be related to the visit of the patient to Indonesia. To investigate whether this species was derived from the environment of the patient, stool samples from both cat and partner were requested. The cat of the patient showed to be positive for *C. helveticus* and not the source for any of the detected species in the patient. The partner and the patient shared a common strain (*C. mucosalis/C. pinnepediorum*), but the route of transmission or a common exposure remained unsolved.

The isolation of *candidatus C. infans* allowed us to perform an antimicrobial susceptibility test and to determine MICs for this strain. The clinical breakpoints (CBP) for these rare species are unknown and therefore we used the CBP of *C. jejuni*. The clinician was able to start an informed treatment instead of empiric treatment of *Campylobacter* species. However, this did not answer the question about causality, as after treatment with tetracycline the patient had ongoing gastrointestinal complaints and was qPCR positive for *Campylobacter* species. Still, *candidatus C. infans* was not detected in the stool sample after treatment with tetracycline.

In conclusion, this study shows the challenges in detection and interpretation of diagnostic results regarding *Campylobacter*. The uncertainty about the relevance of the findings for the clinical condition of the patient remained throughout the case study. The diversity in detected species in this immunocompromised patient with a lack of repeatedly consistent findings resulted in the conclusion that not any of the *Campylobacter* species was the primary cause of the clinical condition. With the lack of proper case-control studies for rare *Campylobacter* species, definite conclusions for the role of these species cannot be made. Given the sporadic detection and the specific condition of the immunocompromised host, no evidence-based guidelines are to be expected for the treatment of non-coli/non-jejuni *Campylobacter* species. As a specific outcome of this study, *candidatus Campylobacter infans* has been isolated for the first time and is currently included in studies in low- and middle-income countries to confirm metagenomics data by culture.

## Figures and Tables

**Figure 1 pathogens-09-01003-f001:**
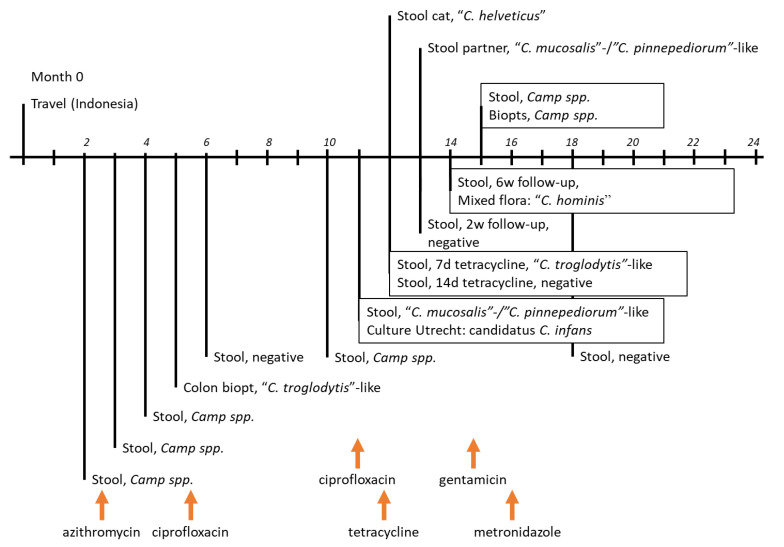
Timeline of the clinical evaluation for patient X. Over the course of two years, several clinical samples were tested for gastrointestinal pathogens. Here we only show the results for the *Campylobacter* 16S rRNA gene qPCR. The timeline represents a period of 2 years, divided in months (italic numbers). Arrows denote antimicrobial treatments; azithromycin, ciprofloxacin, tetracycline, gentamicin and metronidazole.

**Figure 2 pathogens-09-01003-f002:**
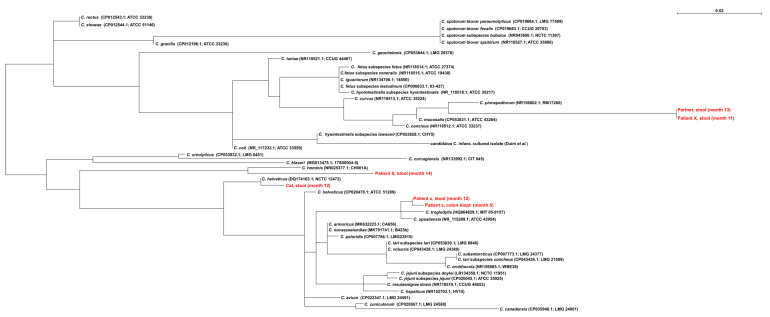
Neighbor-joining phylogenetic tree of a partial 16S rRNA sequence (−470bp) of the *Campylobacter* PCR-positive samples and the cultured *Campylobacter* strain (*candidatus C. infans*) and their homology with reference sequences of several *Campylobacter* species. Clinical samples are marked in red. Reference strains of *Campylobacter* species are given as species (accession number; strain name). Multiple sequence alignment was conducted with ClustalX 2.1 and Seaview version 4.3.3. [19,20]. The bar length indicates the number of nucleotide changes per site.

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
