# Peer review of "A Case of Persistent Diarrhea in a Man with the Molecular Detection of Various Campylobacter species and the First Isolation of candidatus Campylobacter infans"

_pathogens, 2020, doi:10.3390/pathogens9121003_

Round 1

Reviewer 1 Report

The case study describes the identification of multiple Campylobacter species in a patient with persistent diarrhea. However, inconsistent findings in stool samples taken at different times of the study led to the conclusion that none of the detected Campylobacter species was the primary cause of the illness. Moreover, the authors describe the isolation and antibiotic resistance profile of a novel Campylobacter species, “candidatus Campylobacter infans" that was isolated at one time point in this study. Based on their observations and results, the authors conclude that there are challenges in the detection and interpretation of diagnostic results with Campylobacter species making it difficult to develop guidelines for the treatment of this pathogen.

In general:

The introduction provides a good summary of current isolation and detection methods for Campylobacter species and the associated challenges and pitfalls.

The case report itself is well presented with all the necessary details included and the conclusions/interpretations of the findings are convincing.

The authors further discuss the challenges in defining Campylobacter-related disease – since different species have been found at different time points, none could be identified as the causative agent. The authors further highlight that the presence of different species is a diagnostic challenge. However, using the diverse methods presented here, the authors identified an unrecognized species, “candidatus Campylobacter infans”.

Questions/comments:

I am wondering why only microaerobic conditions were used to isolate the Campylobacters? Some Campylobacter species require anaerobic conditions for growth and so if present, these species might have been missed? Maybe the authors can clarify why anaerobic growth conditions were not used (in addition to microaerobic conditions).

Lines 146-147: It is not mentioned in the discussion whether the diagnosis of irritable bowel syndrome was correct? And whether the patient resolved their diarrhea after one course of metronidazole or whether the diarrhea persisted even after the stools were Campylobacter negative?

Line 214: Can this phrase be re-worded for clarity “evidence-based guidelines for treatment are not be expected”.

In Figure 2, can the Patient X samples be in bold or colour so they are more noticeable in the figure?

Spelling:

Line 25: months should be month

Line 30: the partner and the cat

Line 42: The most frequently reported

Line 118: Here, and in the legend to Figure 2, and elsewhere, Campylobacter should be in italics when followed with the word species (also lines 126, 143 and 205)

Line 163: Due to lack of antimicrobial susceptibility data

Line 168: persistent carriage

Line 169: from diarrheal patients

Line 176: as was shown in this study

Line 187: for the other samples

Line 201: remove: was it possible

Line 206: in the stool

Line 212: for the role

Author Response

Questions/comments:

I am wondering why only microaerobic conditions were used to isolate the Campylobacters? Some Campylobacter species require anaerobic conditions for growth and so if present, these species might have been missed? Maybe the authors can clarify why anaerobic growth conditions were not used (in addition to microaerobic conditions). At the Isala hospital, routine culture conditions are optimized for most frequently diagnosed medically important Campylobacter species.

Lines 146-147: It is not mentioned in the discussion whether the diagnosis of irritable bowel syndrome was correct? And whether the patient resolved their diarrhea after one course of metronidazole or whether the diarrhea persisted even after the stools were Campylobacter negative? In line 152 – 154, we describe how the patient was requested to keep a diary regarding his abdominal discomfort, as well as the frequency and appearance of this stools. While the symptoms varied, no correlation was found between the outcome of the Campylobacter PCR (positive/negative) and the stools (regular and painless vs pains and frequent (watery) stools).

Line 214: Can this phrase be re-worded for clarity “evidence-based guidelines for treatment are not be expected”.  We rephrased it to: “Given the sporadic detection and the specific condition of the immunocompromised host, no evidence-based guidelines are to be expected for the treatment of non-coli/non-jejuni Campylobacter species.”

In Figure 2, can the Patient X samples be in bold or colour so they are more noticeable in the figure? Thank you for the suggestion. We have now marked the clinical samples of Patient X, the partner and the cat in red and bigger font size.

Spelling: Thank you very much. We have altered the text as suggested below.

Line 25: months should be month

Line 30: the partner and the cat

Line 42: The most frequently reported

Line 118: Here, and in the legend to Figure 2, and elsewhere, Campylobacter should be in italics when followed with the word species (also lines 126, 143 and 205)

Line 163: Due to lack of antimicrobial susceptibility data

Line 168: persistent carriage

Line 169: from diarrheal patients

Line 176: as was shown in this study

Line 187: for the other samples

Line 201: remove: was it possible

Line 206: in the stool

Line 212: for the role

Reviewer 2 Report

Dr. Flipse and colleagues present an exciting and well-written case report that contributes significantly to the field. 

Before acceptance, the following rather minor issues should be addressed, however:

line 22: A verb is missing in this first sentence of the abstract: rephrase or delete "who"?!

line 25ff: all abbreviations including species names need to be introduced upon first appearance and written on full.

line 46: references need to be included for the second part of the sentence.

line 56: "have their limitations inn sensitivity": What about specificity? 

line 64: "venereum"

line 79ff: for all media include vendor/company, city, country

line 87: months 3 and 4

line 99ff: "Campylobacter" in italics throughout the text including figure legends

line 104: introduce "2dd" upon first appearance (not everybody is familiar with this abbreviation/order, particularly non-clinicians)

Author Response

Before acceptance, the following rather minor issues should be addressed, however:

line 22: A verb is missing in this first sentence of the abstract: rephrase or delete "who"?! “Who” was deleted.

line 25ff: all abbreviations including species names need to be introduced upon first appearance and written on full. Thank you, we screened the text again.

line 46: references need to be included for the second part of the sentence. True, we moved the references to the end of the sentence.

line 56: "have their limitations inn sensitivity": What about specificity? We stressed that both sensitivity and specificity can be limited based on the assay used.

line 64: "venereum" Thank you.

line 79ff: for all media include vendor/company, city, country We added this information.

line 87: months 3 and 4 thank you.

line 99ff: "Campylobacter" in italics throughout the text including figure legends. Done.

line 104: introduce "2dd" upon first appearance (not everybody is familiar with this abbreviation/order, particularly non-clinicians) Done.

Reviewer 3 Report

This is an interesting case study depicting the challenges in detecting and interpreting Campylobacter diagnostic results.  It highlights the difficulties in being able to determine causation and drawing clinical conclusions. It also highlights the advantages of using molecular assays but also its limitations. A key outcome of this exploration is the isolation of candidatus Campylobacter infans for the first time, triggering further studies to be able to confirm the metagenomics data in LMIC.  Overall it's a potentially useful case for microbiologists and clinicians embarking on a  Campylobacter trail.  Timeline and phylogenetic tree both very helpful for understanding the details of the exploration.

Author Response

Thank you for your comments.